# CDK4/6-dependent activation of DUB3 regulates cancer metastasis through SNAIL1

Tongzheng Liu[1,2], Jia Yu[3], Min Deng[2], Yujiao Yin[4,5], Haoxing Zhang[6], Kuntian Luo[2,4,5], Bo Qin[2], Yunhui Li[4,5], Chenming Wu[4,5], Tao Ren[7], Yang Han[8], Peng Yin[9], JungJin Kim[2], SeungBaek Lee[2], Jing Lin[10], Lizhi Zhang[11], Jun Zhang[11], Somaira Nowsheen[12], Liewei Wang[3], Judy Boughey[13], Matthew P. Goetz[2,3], Jian Yuan[2,4,5] & Zhenkun Lou[2]

Tumour metastasis, the spread of cancer cells from the original tumour site followed by growth of secondary tumours at distant organs, is the primary cause of cancer-related deaths and remains poorly understood. Here we demonstrate that inhibition of CDK4/6 blocks breast tumour metastasis in the triple-negative breast cancer model, without affecting tumour growth. Mechanistically, we identify a deubiquitinase, DUB3, as a target of CDK4/6; CDK4/6-mediated activation of DUB3 is essential to deubiquitinate and stabilize SNAIL1, a key factor promoting epithelial–mesenchymal transition and breast cancer metastasis. Overall, our study establishes the CDK4/6–DUB3 axis as an important regulatory mechanism of breast cancer metastasis and provides a rationale for potential therapeutic interventions in the treatment of breast cancer metastasis.

[1] Jinan University Institute of Tumor Pharmacology, Guangzhou 510632, China. [2] Department of Oncology, Mayo Clinic, Rochester, Minnesota 55905, USA. [3] Department of Molecular Pharmacology and Experimental Therapeutics, Mayo Clinic, Rochester, Minnesota 55905, USA. [4] Research Center for Translational Medicine, East Hospital, Tongji University School of Medicine, Shanghai 200120, China. [5] Key Laboratory of Arrhythmias of the Ministry of Education of China, East Hospital, Tongji University School of Medicine, Shanghai 200120, China. [6] School of Life Sciences, Southwest University, Chongqing 400715, China. [7] Department of Respiratory Medicine, East Hospital, Tongji University School of Medicine, Shanghai 200120, China. [8] Department of Pathology, East Hospital, Tongji University School of Medicine, Shanghai 200120, China. [9] Department of Immunology, Mayo Clinic, Rochester, Minnesota 55905, USA. [10] First Affiliated Hospital, Chinese PLA General Hospital, Beijing 100818, China. [11] Department of Laboratory Medicine and Pathology, Mayo Clinic, Rochester, Minnesota 55905, USA. [12] Medical Scientist Training Program, Mayo Medical School and Mayo Graduate School, Mayo Clinic, Rochester, Minnesota 55905, USA. [13] Division of Subspecialty General Surgery, Mayo Clinic, Rochester, Minnesota 55905, USA. Correspondence and requests for materials should be addressed to J.Y. (email: Yuanjian229@hotmail.com) or to Z.L. (email: Lou.zhenkun@Mayo.edu).

Tumour metastasis, the spread of cancer cells to distant vital organs, causes the majority of human cancer-related deaths[1–6]. Metastasis occurs through a complex multistep process including local invasion, intravasation, transport, extravasation and colonization, which requires the concerted action of many genes and signal pathways[7–13]. Epithelial–mesenchymal transition (EMT) is a highly conserved cellular process in which polarized, immobile epithelial cells are converted to migratory mesenchymal cells[14]; EMT has been accepted as a critical process during embryogenesis[15,16]. Numerous studies have clearly demonstrated the importance of EMT in tissue regeneration, tumour progression and metastasis[13,16–19], while some studies suggest a major role of EMT in chemoresistance[20–24]. Loss of the cell adhesion molecule E-cadherin is a functional event of EMT that reinforces the destabilization of adherent junctions. Loss of E-cadherin and other essential events in the EMT process alter the phenotype of epithelial cells from noninvasive to invasive[16,25]. Downregulation of E-cadherin involves several mechanisms, such as transcriptional repression, mutation and increased cleavage or degradation of E-cadherin[1,16]. The transcription factor SNAIL1 directly represses the expression of *CDH1*, the gene encoding E-cadherin, and activates the expression of invasion-associated genes, thereby promoting EMT[26–28]. For example, Peinado et al.[29] demonstrated that SNAIL1 interacts with a co-repressor complex SIN3A, and histone deacetylases HDAC1 and HDAC2, to repress E-cadherin expression by modifying local chromatin structure. SNAIL1 also interacts with G9a, a major euchromatin methyltransferase responsible for H3K9me2, and recruits G9a and DNA methyltransferases to the E-cadherin promoter for DNA methylation[30].

Expression of SNAIL1 is tightly regulated at the transcriptional and post-transcriptional levels. TGFβ, NOTCH and WNT pathways can transcriptionally increase SNAIL1 expression by directly binding to the SNAIL1 promoter, thus inducing EMT in mammalian cells[28,31,32]. As a highly unstable protein, SNAIL1 is post-translationally degraded through the ubiquitin-proteasome pathway. Several E3 ubiquitin ligases, including FBXL14, FBXW1 and FBXO11, have been shown to induce SNAIL1 ubiquitination and degradation[33–35]. However, whether and how SNAIL1 is stabilized post-translationally is unclear. Thus, the identification of the signalling pathway controlling SNAIL1 stabilization will provide important clues for potential therapeutic targets of cancer metastasis. Here we demonstrate CDK4/6 inhibition blocks breast tumour metastasis without affecting tumour growth. Furthermore, we identify deubiquitinating enzyme 3 (DUB3) as a new target of CDK4/6; CDK4/6-mediated activation in DUB3 is essential to deubiquitinate and stabilize SNAIL1. Overall, our study demonstrates the importance of CDK4/6–DUB3 axis in regulating breast cancer metastasis and provides a rationale for potential therapeutic interventions in the treatment of breast cancer metastasis.

## Results

### CDK4/6 inhibitor PD0332991 decreases human cancer metastasis.
A specific inhibitor of CDK4/6, PD0332991 (palbociclib), is a potent anti-proliferative agent in multiple cancer cells[36–38]. More recently, the US Food and Drug Administration granted accelerated approval for palbociclib for use in combination with letrozole for the treatment of postmenopausal women with oestrogen receptor (ER)-positive, HER2-negative advanced breast cancers[39,40]. Tumours that are ER negative, progesterone receptor negative and HER2 negative, also known as triple-negative breast cancers (TNBCs), usually grow the fastest and are more metastatic. In a previous study, compared with ER-positive breast cancer lines, TNBCs were more resistant to PD0332991

(ref. 36). In our study, PD0332991 treatment inhibited cell proliferation of T47D cells, but had no effect on cell proliferation of TNBC cell lines, BT-549 and MDA-MB-231 (Supplementary Fig. 1a–c). Surprisingly, PD0332991 treatment markedly reduced cell migration of MDA-MB-231 *in vitro* (Fig. 1a), suggesting a potential role of CDK4/6 inhibitor in blocking cell metastasis. PD0332991 was further tested in a patient-derived xenograft model generated from a triple-negative high-grade invasive ductal carcinoma (from the Breast Cancer Genome-Guided Therapy study (BEAUTY) in Mayo Clinic in Rochester, Minnesota). Immunodeficient mice implanted with human breast tumour biopsy sample HCI001 were found to have liver, lung and ovary metastases, reflecting the metastatic pattern in the donor patient. Administration of PD0332991 did not affect the growth of primary tumour as shown in Fig. 1b and Supplementary Fig. 1d. Strikingly, we found that PD0332991 could significantly decrease liver (12.5% versus 75%) and lung (25% versus 75%) metastases compared with the saline group (Fig. 1c–e,f–h). We also investigated PD0332991 function in a xenograft metastasis model. MDA-MB-231 cells were injected into the mammary fat pad of immunodeficient mice. When tumours reached 400 mm$^3$ in size, we removed the primary tumours and treated these mice with either vehicle or PD0332991 for an additional 12 weeks to examine lung metastases. We found that the administration of PD0332991 markedly suppressed lung colonization in these mice, as determined by the number of metastatic lung nodules (Supplementary Fig. 1e,f). These results demonstrate that the CDK4/6 inhibitor PD0332991 can inhibit TNBC metastasis without affecting tumour growth in the two models we tested. This finding could potentially broaden the use of CDK4/6 inhibitor in the treatment of TNBC metastasis.

Numerous studies have clearly demonstrated that EMT is essential for tumour progression and metastasis. As shown in Fig. 1i, PD0332991 treatment decreased vimentin (mesenchymal marker) protein level, while increasing E-cadherin (epithelial marker) protein level in MDA-MB-231. Interestingly, the protein level of SNAIL1, a key factor in EMT, decreased upon PD0332991 treatment (Fig. 1i), which could be rescued by the proteasome inhibitor MG-132 (Fig. 1j,k). These results suggest that CDK4/6 inhibition regulates SNAIL1 protein level in a proteasome dependent manner. Furthermore, PD0332991 treatment markedly decreased the SNAIL1 protein stability (Fig. 1l,m). PD0332991 treatment increased the ubiquitination of SNAIL1 (Fig. 1n,o). Previous studies suggested that several E3 ubiquitin ligases including FBXL14, FBXW1 and FBXO11 regulate SNAIL1 ubiquitination and degradation[33–35]. Neither the direct interaction between CDK4/6 and SNAIL1 (Supplementary Fig. 1g) nor CDK4/6-mediated phosphorylation of SNAIL1 were detected (Supplementary Fig. 1h). These results indicate a potential unidentified factor mediates CDK4/6's regulation of SNAIL1.

**Identification of deubiquitinases for SNAIL1**. To identify the potential linker between CDK4/6 and SNAIL1, we used MDA-MB-231 cells stably expressing FLAG-SNAIL1 to perform tandem affinity purification and mass spectrometry analysis. In addition to known SNAIL1 interacting proteins such as ATM, DNMT1 and CSNK2A1 (refs 30,41,42), we identified several DUBs, including USP7, USP10, USP11 and DUB3, as major SNAIL1-associated proteins (Fig. 2a). Although SNAIL1 interacted with several DUBs (Supplementary Fig. 2a–d), only overexpression of DUB3, but not USP7, USP10 or USP11, markedly increased the protein level of SNAIL1 in MDA-MB-231 (Fig. 2b). We confirmed the endogenous SNAIL1–DUB3 interaction by co-immunoprecipitation in both MDA-MB-231 and BT-549 cells (Fig. 2c,d; Supplementary Fig. 2e,f). As shown in Fig. 2c and

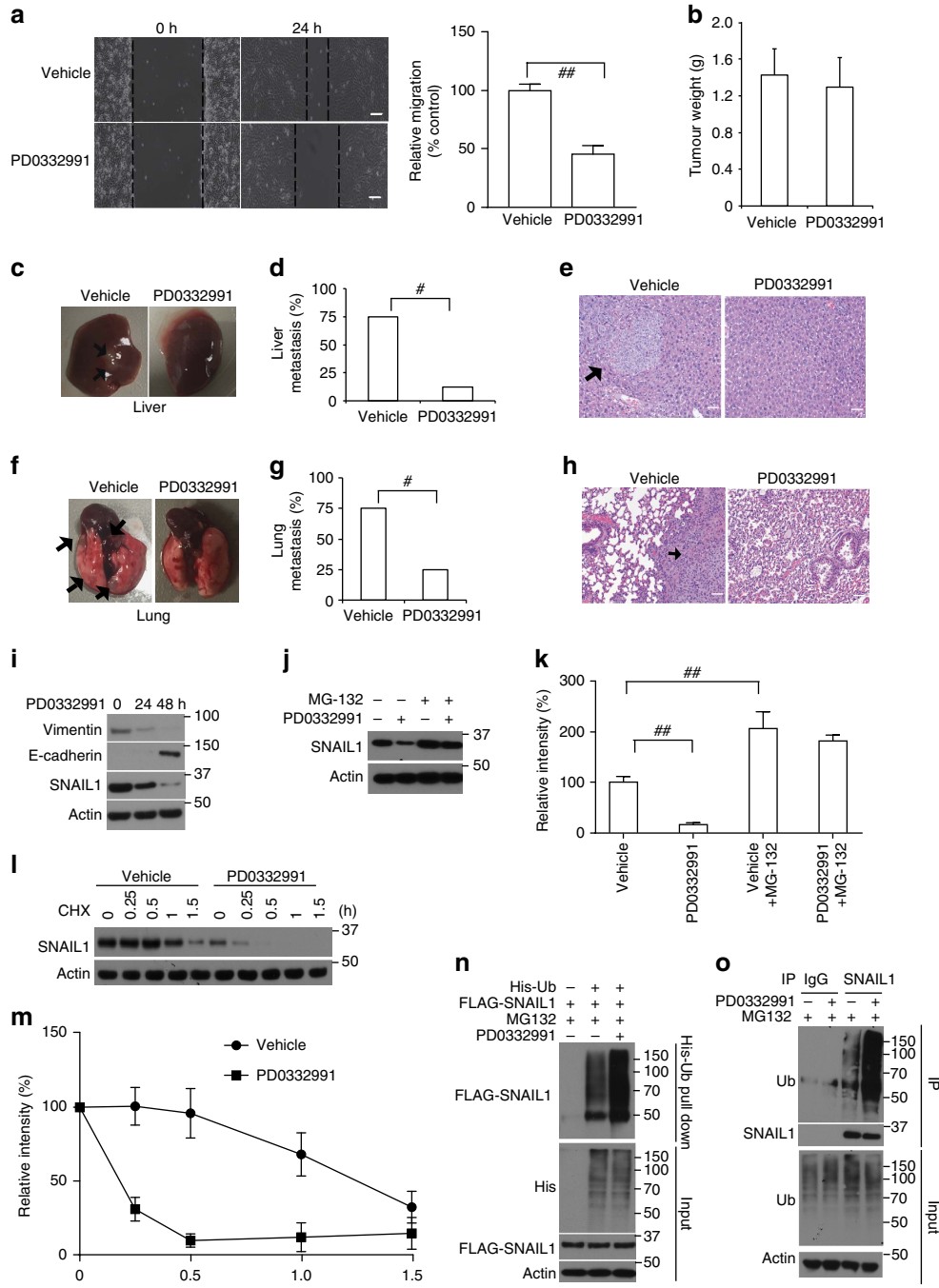

**Figure 1 | CDK4/6 inhibition inhibits cancer metastasis. (a)** MDA-MB-231 cells were treated with PD0332991 as indicated. The migratory potential of cells was analysed by wound-healing assay and results were quantified (right panel). Scale bars, 100 μm. The results represent the means ± s.d. of three independent experiments. $^{\#\#}P<0.01$. (**b–h**) Passage 3 tumours from HCI001 were used to test the effect of PD0332991 on metastasis. When primary tumours reached 100–150 mm³, mice were randomized and treated either with saline or PD0332991 for 5 weeks ($n=8$). Tumour weights (**b**) were measured after mice were killed. Data are expressed as mean ± s.d. Statistical analyses were performed with the Student's $t$-test. Liver (**c–e**) and lung (**f–h**) metastatic nodules were examined macroscopically or detected in paraffin-embedded sections stained with H&E. Scale bars, 50 μm. Arrowheads indicate metastases. Fisher's exact test was calculated and statistical significance is represented ($^{\#}P<0.05$). (**i**) MDA-MB-231 cells were treated with PD0332991 and western blot was performed with indicated antibodies. (**j,k**) MDA-MB-231 cells were treated with PD0332991 for 24 h. Then, cells were treated with either vehicle or MG-132 for an additional 6 h. SNAIL1 level was detected by western blotting. Three independent experiments were performed and results are quantified in (**k**). $^{\#\#}P<0.01$. (**l,m**) MDA-MB-231 cells were treated with PD0332991 for 24 h. Cycloheximide pulse-chase assay was performed in cells. Three independent experiments were performed and results are quantified in **m**. (**n**) MDA-MB-231 cells were transfected with indicated constructs and treated with vehicle or PD0332991 for 24 h in the presence of MG-132. Ni-NTA beads were used to pull down His-tagged ubiquitin, and the polyubiquitylated SNAIL1 protein was examined. (**o**) MDA-MB-231 cells were treated vehicle or PD0332991 for 24 h in the presence of MG-132 and cell lysates were subjected to immunoprecipitation with control IgG, anti-SNAIL1 and the ubiquitination of SNAIL1 protein was examined.

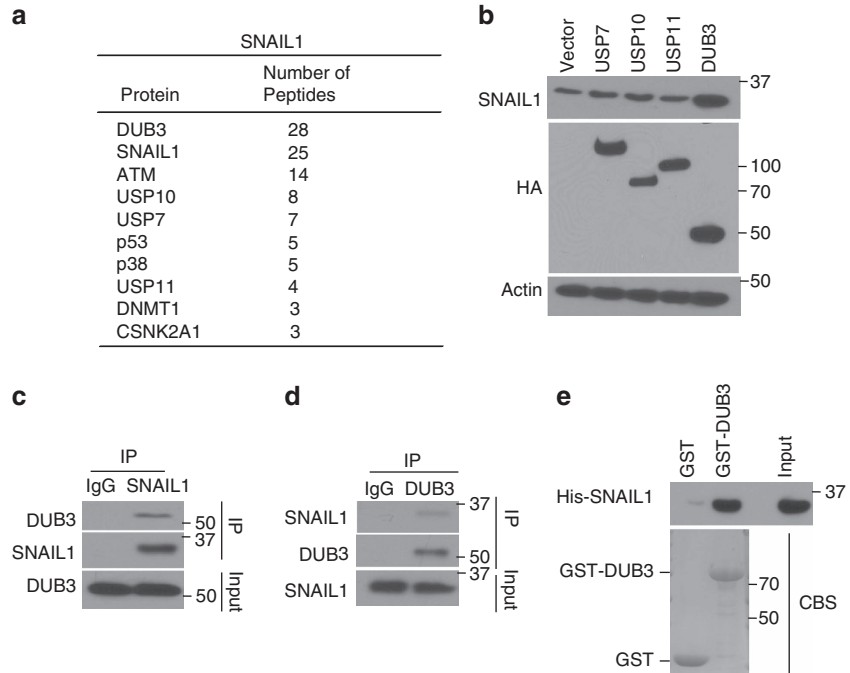

**Figure 2 | DUB3 interacts with SNAIL1.** (**a**) List of SNAIL1-associated proteins identified by mass spectrometric analysis. MDA-MB-231 cells stably expressing FLAG-SNAIL1 were generated and SNAIL1 complexes were subjected to mass spectrometric analysis. (**b**) MDA-MB-231 cells were transfected with indicated plasmids and western blotting was performed. (**c**,**d**) MDA-MB-231 cell lysates were subjected to immunoprecipitation with control IgG, anti-SNAIL1 (**c**) or anti-DUB3 (**d**) antibodies. The immunoprecipitates were then blotted with the indicated antibodies. (**e**) Purified recombinant GST, GST-DUB3 and His-SNAIL1 were incubated *in vitro* as indicated. The interaction between DUB3 and SNAIL1 was then examined. CBS, Coomassie blue staining.

Supplementary Fig. 2e, SNAIL1 co-immunoprecipitated with DUB3. Reciprocal immunoprecipitation with DUB3 antibodies also brought down SNAIL1 (Fig. 2d; Supplementary Fig. 2f). Moreover, GST-DUB3 could interact with recombinant His-SNAIL1 *in vitro*, indicating a direct interaction between DUB3 and SNAIL1 (Fig. 2e).

**DUB3 deubiquitinates and stabilizes SNAIL1**. The interaction of DUB3 and SNAIL1 prompted us to examine a potential role for DUB3 in the regulation of SNAIL1 stability and function. First, DUB3 and SNAIL1 protein levels were examined in several luminal- and basal-like breast cancer cell lines. As shown in Fig. 3a, DUB3 and SNAIL1 protein levels are higher in basal-like breast cancer cell lines. To directly test the function of DUB3 on endogenous SNAIL1 protein stability, we knocked down DUB3 with its specific short hairpin RNAs (shRNAs) in MDA-MB-231. Depletion of DUB3 significantly decreased SNAIL1 protein level (Fig. 3b). The regulation of SNAIL1 protein stability by DUB3 was not at the level of transcription since no apparent difference of SNAIL1 mRNA level was detected in cells stably expressing control and DUB3 shRNAs (Fig. 3c). On the other hand, MG-132 treatment could rescue the decreased SNAIL1 protein level in cells depleted of DUB3 (Fig. 3d). Moreover, overexpression of wild-type (WT) DUB3, but not the catalytically inactive C89S mutant in both MCF-7 and T47D cells, increased SNAIL1 protein level (Fig. 3e). Furthermore, SNAIL1 protein was less stable in cells depleted of DUB3 in a cycloheximide pulse-chase assay (Fig. 3f,g). These results suggest that DUB3 regulates SNAIL1 stability. To further investigate whether DUB3 functions as a bona fide DUB that deubiquitinates SNAIL1, we performed a deubiquitination assay by cotransfecting cells with WT DUB3 or the C89S mutant. A significant decrease of polyubiquitylated SNAIL1 protein was observed in cells transfected with WT

DUB3, whereas the expression of C89S mutant was not able to decrease SNAIL1 ubiquitination (Fig. 3h). In addition, WT DUB3, but not the C89S mutant, markedly decreased SNAIL1 ubiquitination *in vitro* (Fig. 3i). On the other hand, depletion of DUB3 significantly increased SNAIL1 ubiquitination (Fig. 3j; Supplementary Fig. 3a). When we examined the linkage of SNAIL1 ubiquitination, we found that SNAIL1 was ubiquitinated through both K48- and K63-specific chains (Supplementary Fig. 3b). In addition, we found that DUB3 regulates only K48, but not K63 ubiquitin chains (Supplementary Fig. 3c,d). Taken together, our results suggest that DUB3 is a bona fide DUB targeting SNAIL1 protein for deubiquitination and stabilization.

**DUB3 regulates EMT through SNAIL1**. Accumulating experimental and clinical evidences demonstrated that SNAIL1 promotes EMT. To investigate the potential function of DUB3 in this process, we overexpressed DUB3 in two luminal breast cancer cell lines, MCF-7 and T47D. Expression of WT DUB3 resulted in decreased E-cadherin (epithelial marker) expression, increased vimentin (mesenchymal marker) expression and converted luminal cells into basal-like phenotype (Fig. 4a,b). Furthermore, the effect of DUB3 overexpression on EMT was not observed in cells transfected with SNAIL1 shRNA, suggesting that DUB3 functions in promoting basal-like phenotype conversion through SNAIL1 (Fig. 4a,b). Collectively, these results suggest that DUB3 regulates EMT by stabilizing SNAIL1.

**DUB3 regulates cell migration and cancer metastasis**. Repression of E-cadherin expression by SNAIL1 or other EMT factors is one of the critical events in EMT induction and cancer metastasis[1,28,43]. Since DUB3 deubiquitinates and stabilizes SNAIL1 and consequently decreases E-cadherin expression, we hypothesized that DUB3 is critical for breast cancer cell migration

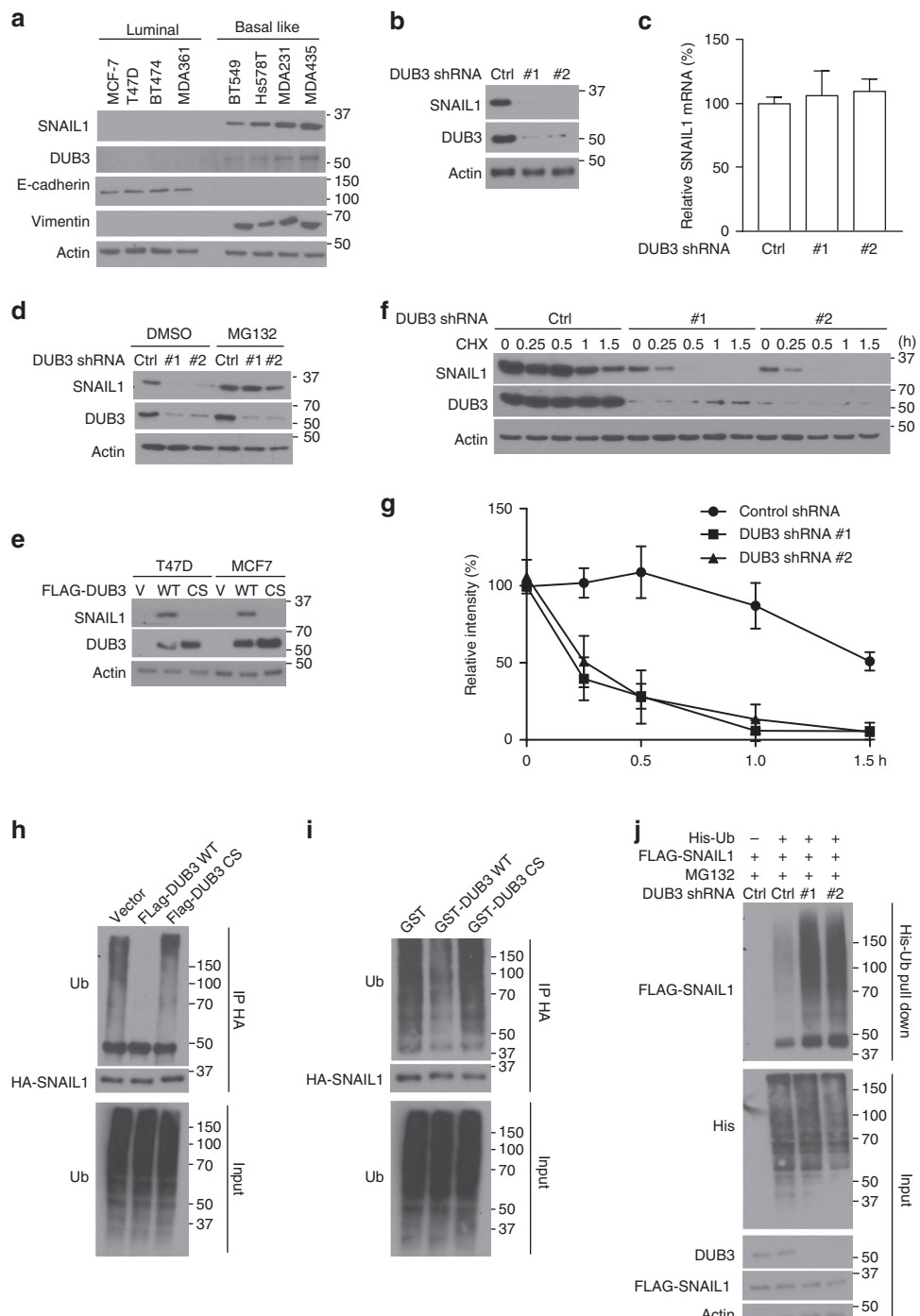

**Figure 3 | DUB3 deubiquitinates and stabilizes SNAIL1.** (**a**) Cell extracts were prepared from four luminal- and four basal-like subtypes of human breast cancer cell lines, and expression of SNAIL1, DUB3, E-cadherin and vimentin was analysed by western blotting. (**b**) MDA-MB-231 cells stably expressing control or DUB3 shRNAs were generated and western blot was performed with the indicated antibodies. (**c**) Total RNA was isolated from cells in **b**. Relative expression of SNAIL1 in cells stably expressing control or DUB3 shRNAs was determined by quantitative PCR. Transcript levels were determined relative to actin mRNA levels and normalized relative to control cells. The results represent the means ( ± s.d.) of three independent experiments. (**d**) MDA-MB-231 cells stably expressing control or DUB3 shRNAs were treated with vehicle or MG-132 and western blot was performed with the indicated antibodies. (**e**) T47D and MCF-7 were infected with virus containing vector, FLAG-DUB3 and the C89S mutant and western blot was performed. (**f**) Cycloheximide pulse-chase assay was performed in cells as in **b** and results are quantified in **g**. The results represent the means ( ± s.d.) of three independent experiments. $^{\#\#}P < 0.01$. (**h**) Cells were cotransfected with indicated plasmids and treated with MG-132 for 6 h before cell lysates were boiled and immunoprecipitated with HA beads, and the polyubiquitylated SNAIL1 protein was detected by anti-ubiquitin antibody. (**i**) Cells were transfected with HA-SNAIL1 and treated with MG-132 for 6 h. Cell lysates were immunoprecipitated with HA beads and incubated with GST, GST-DUB3 or GST-DUB3 C89S mutant in a cell-free condition. The polyubiquitylated SNAIL1 protein was detected by anti-ubiquitin antibody. (**j**) MDA-MB-231 cells stably expressing control or DUB3 shRNAs were transfected with indicated constructs and Ni-NTA beads were used to pull down His-tagged ubiquitin, and the polyubiquitylated SNAIL1 protein was examined.

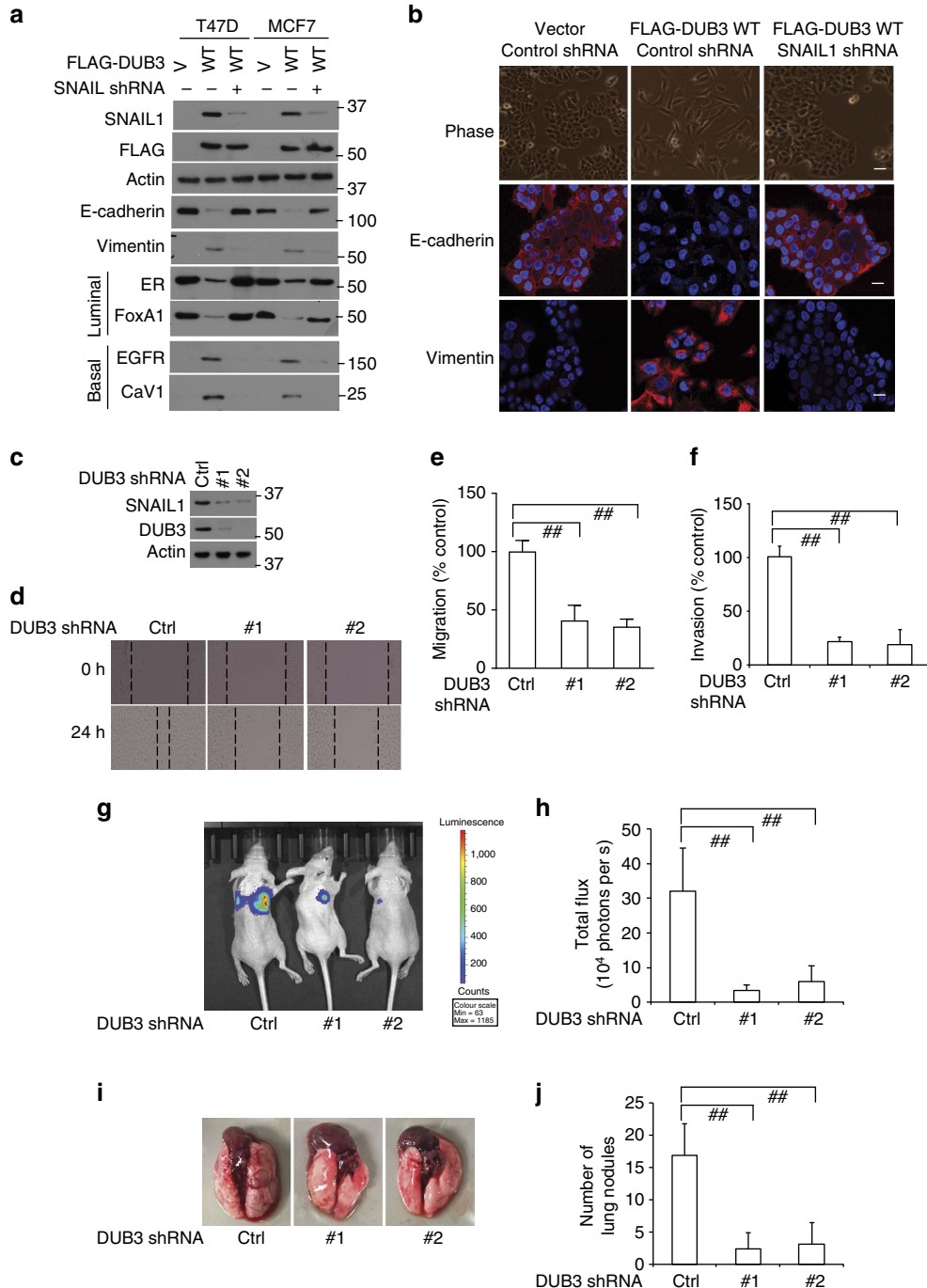

**Figure 4 | DUB3 regulates EMT through SNAIL1. (a)** T47D and MCF-7 cells were transfected with indicated plasmids and western blot was performed with indicated antibodies. **(b)** T47D cells were transfected with indicated plasmids and cell morphological changes associated with EMT are shown in the phase contrast images. Expression of E-cadherin and vimentin was analysed by immunofluorescence. Nuclei were visualized with 4,6-diamidino-2-phenylindole staining (blue). Scale bars, 25 μm. **(c–e)** MDA-MB-231 cells were stably transfected with control and DUB3 shRNAs **(c)**. **(d)** The migratory ability of cells was analysed by wound-healing assay and results are quantified in **e**. The results represent the means ± s.d. of three independent experiments. $^{\#\#}P < 0.01$. **(f)** The invasiveness of cells was analysed with a chamber invasion assay. The results represent the means ( ± s.d.) of three independent experiments. $^{\#\#}P < 0.01$. **(g–j)** One million cells from **c** were injected into the lateral tail vein of immunodeficient mice ($n = 8$). After 6 weeks, the development of lung metastases was recorded using **(g)** bioluminescence imaging and **(h)** quantified. After 12 weeks, mice were killed and lung metastatic nodules were counted and quantified **(i,j)**. Data are expressed as the means ± s.d. Statistical analyses were performed with the analysis of varinace. $^{\#\#}P < 0.01$.

and invasion *in vitro*. Depletion of DUB3 significantly inhibited the migratory ability and invasiveness of MDA-MB-231 cells (Fig. 4c–f), although it did not affect cell proliferation (Supplementary Fig. 4a,b). However, overexpression of SNAIL1 in cells rescued the decreased migratory ability induced by depletion of DUB3 (Supplementary Fig. 4c–e). Altogether, these results indicate that SNAIL1 is an important factor for DUB3's effect on migration.

We further investigated DUB3 function in a xenograft metastasis model. We found that knockdown of DUB3 in MDA-MB-231 cells markedly suppresses lung colonization in these mice, as determined by intensity of bioluminescence (Fig. 4g,h) and number of lung nodules present (Fig. 4i,j). Furthermore, overexpression of SNAIL1 in cells with DUB3 depletion significantly increased lung colonization compared with those depleted of DUB3, as determined by intensity of bioluminescence (Supplementary Fig. 4f,g) and number of lung nodules present (Supplementary Fig. 4h). Together, our results demonstrate that DUB3 is critical for cell migration, invasion and lung colonization through stabilizing SNAIL1.

**DUB3 is positively correlated to SNAIL1 in breast cancers**. Since SNAIL1 plays a critical role in human breast carcinoma metastasis[33,43,44] and DUB3 stabilizes SNAIL1 by deubiquitinating SNAIL1, it is possible that DUB3 also facilitates breast carcinoma metastasis. Our results (shown in Fig. 4 and Supplementary Fig. 4) demonstrate DUB3's ability to increase breast cancer cell migration and metastasis by targeting SNAIL1, thereby supporting our hypothesis that DUB3 promotes breast carcinoma metastasis in patients. To further test this hypothesis, we examined the expression of DUB3 and SNAIL1 in breast cancer tissue samples by using breast cancer tissue microarray. Notably, high DUB3 protein expression positively correlated with metastatic carcinoma ($P < 0.0001$, $R = 0.347$) (Supplementary Fig. 5a,b). High SNAIL1 protein expression was also positively correlated with metastatic carcinoma ($P < 0.0001$, $R = 0.344$; Supplementary Fig. 5a,c). Importantly, DUB3 expression positively correlated with SNAIL1 protein expression in metastatic carcinoma (Supplementary Fig. 5d). In addition, in another tissue array with known breast cancer subtypes, high protein expressions of DUB3 and SNAIL1 were both positively correlated with basal-like breast cancer type (Supplementary Fig. 5e,f), which is consistent with the role of DUB3 in promoting basal-like phenotype conversion we observed in Fig. 4a,b. These results suggest that DUB3 and SNAIL1 are positively correlated in metastatic breast carcinoma.

**CDK4/6 phosphorylates Ser41 of DUB3**. Our results demonstrate that both CDK4/6 and DUB3 regulate EMT and metastasis through stabilizing SNAIL1. We speculated that DUB3 might be the missing link between CDK4/6 and SNAIL1. Interestingly, we found that PD0332991 could not further induce decrease of SNAIL1 protein level in DUB3 knockdown cells, suggesting that CDK4/6 may regulate SNAIL1 in a DUB3-dependent manner (Fig. 5a). The interaction between CDK4/6 and DUB3 was also detected by endogenous co-immunoprecipitation and *in vitro* pull down (Fig. 5b,c). We hypothesized that CDK4/6 could phosphorylate DUB3. Indeed, we found that CDK4/6 could phosphorylate DUB3 *in vitro* (Supplementary Fig. 6a). To further identify the potential phosphorylation sites in DUB3, we generated MDA-MB-231 cells stably expressing FLAG-DUB3 to perform tandem affinity purification and mass spectrometry analysis of potential phosphorylation events in DUB3. We found that Ser41 is a major phosphorylation site on DUB3, which matches with a CDK consensus motif (Supplementary Fig. 6b). To test whether S41 is a CDK4/6 phosphorylation site, GST-fused WT DUB3 and S41A mutants were incubated with active CDK4 or CDK6 and an *in vitro* kinase assay was performed. As shown in Fig. 5d, mutation at S41 abolished CDK4/6-mediated phosphorylation of DUB3 *in vitro*. A phospho-Ser41-specific antibody was generated to further study the phosphorylation of Ser41 in cells. HEK293T cells were transfected with DUB3 WT or the S41A

mutant. As shown in Fig. 5e, WT DUB3 was phosphorylated in cells. However, S41A mutation completely abrogated the phosphorylation of DUB3 at this site, indicating the specificity of this antibody. We next tested CDK4/6-mediated phosphorylation of DUB3 in cells. We found that CDK4/6 inhibition markedly reduces the phosphorylation of DUB3 at Ser41 (Fig. 5f). Furthermore, depletion of CDK4 or CDK6 in cells only partially decreased the phosphorylation of DUB3, while double knockdown of CDK4 and CDK6 almost completely abrogated DUB3 phosphorylation (Fig. 5g). In addition, depletion of CDK4 or CDK6 alone in cells only partially decreased SNAIL1, while double knockdown of CDK4 and CDK6 decreased SNAIL1 more markedly (Fig. 5h). Interestingly, CDK1 also phosphorylated DUB3 at Ser41 *in vitro* and the treatment of Roscovitine could decrease the phosphorylation of DUB3 (Supplementary Fig. 7a,b). Thus, the potential function of CDK1 inhibition on EMT and breast cancer metastasis warrants further investigation. To assess whether PD0332991 actions are mediated through CDK4/6–DUB3–SNAIL1 axis, we expressed WT DUB3, S41A or S41D mutants in cells depleted of endogenous DUB3 and then treated these cells with either vehicle or PD0332991. We found that PD0332991 induces decrease in SNAIL1 protein level only in WT DUB3 reconstituted cells but not in cells reconstituted with S41A or S41D mutants (Fig. 5i). These findings provide evidence that Ser41 is the major CDK4/6-mediated phosphorylation site on DUB3, which in turn regulates SNAIL1 protein level.

**CDK4/6 phosphorylates and activates DUB3**. Next, we investigated whether CDK4/6-mediated phosphorylation affects DUB3 activity. The deubiquitinase activity of WT DUB3 and S41A or S41D mutants towards fluorogenic substrate, Ub-AMC, were measured. Compared with the S41A mutant, WT and the S41D mutant showed much higher activity towards Ub-AMC (Supplementary Fig. 8a,b). This result suggests that S41 phosphorylation is essential for the deubiquitinase activity of DUB3. We next tested how CDK4/6 and S41 phosphorylation of DUB3 affects SNAIL1 ubiquitination. We transfected cells with FLAG-DUB3 and HA-SNAIL1, treated cells with vehicle or CDK4/6 inhibitor PD0332991, and SNAIL1 ubiquitination was determined. As shown in Fig. 6a, SNAIL1 ubiquitination was stronger in cells treated with PD0332991 compared to vehicle, suggesting that CDK4/6 activity is important for the catalytic activity of DUB3. We further tested whether phosphorylation of S41 on DUB3 could affect the ubiquitination of SNAIL1 in cells. As shown in Fig. 6b,c, overexpression of both WT and S41D mutant DUB3 in cells efficiently decreased the ubiquitination of SNAIL1; however, S41A mutant failed to do so. Collectively, these results suggest that phosphorylation of S41 is important for the deubiquitinase activity of DUB3 towards SNAIL1. We further tested whether phosphorylation of S41 on DUB3 could affect the protein level of SNAIL1 in cells. We reconstituted shRNA-resistant WT, S41A and S41D DUB3 mutants in cells depleted of endogenous DUB3 and checked SNAIL1 level. As shown in Fig. 6d, DUB3 knockdown efficiently decreased SNAIL1 protein level. WT DUB3 and the S41D mutant could rescue SNAIL1 protein level, while the S41A mutant failed to do so. Moreover, depletion of DUB3 greatly inhibited the migratory ability of MDA-MB-231 cells. Cells reconstituted with WT DUB3 and the S41D mutant, but not the S41A mutant could rescue this phenotype (Fig. 6e,f). Results from the metastasis model showed that the reconstitution of WT DUB3 and the S41D mutant in MDA-MB-231 cells stably expressing DUB3 shRNA causes stronger lung colonization than that of S41A mutant, as determined by the number of lung nodules (Fig. 6g) and by intensity of bioluminescence (Supplementary Fig. 8c,d). These

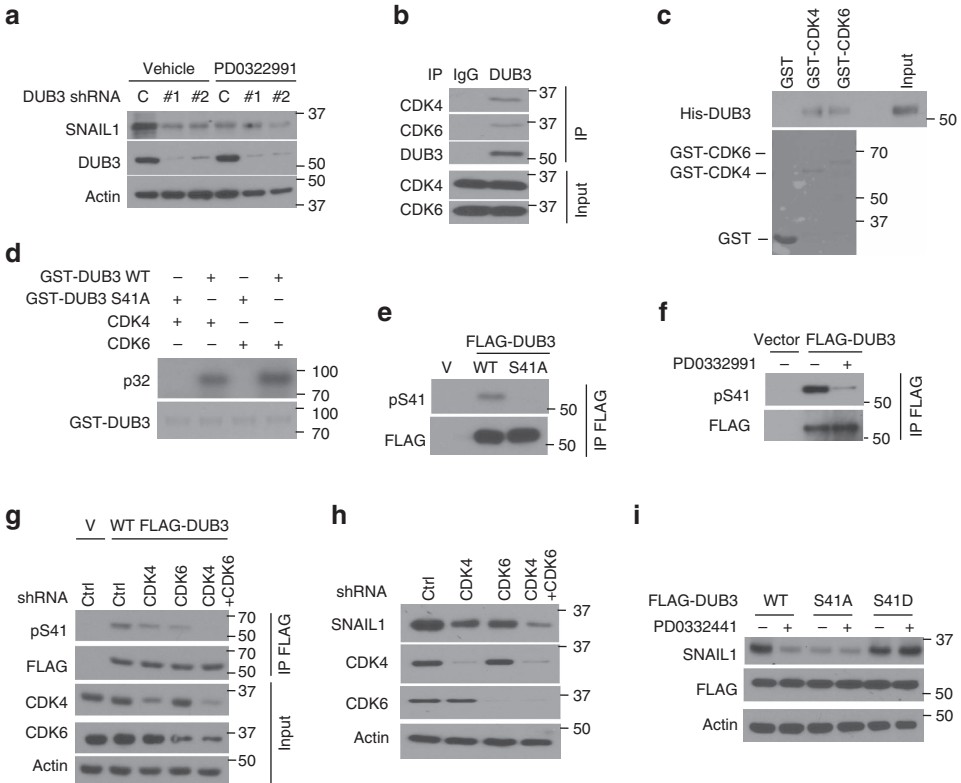

**Figure 5 | CDK4/6 phosphorylates DUB3 at Ser 41.** (**a**) MDA-MB-231 cells stably expressing control or DUB3 shRNAs were treated with either vehicle or PD0332991 for 24 h. Western blot was performed with the indicated antibodies. (**b**) MDA-MB-231 cell lysates were subjected to immunoprecipitation with control IgG or anti-DUB3 antibodies. The immunoprecipitates were then blotted with the indicated antibodies. (**c**) Purified recombinant GST, GST-CDK4, GST-CDK6 and His-DUB3 were incubated *in vitro* as indicated. The interaction between DUB3 and CDK4/6 was then examined. CBS, Coomassie blue staining. (**d**) CDK4/6 phosphorylates DUB3 *in vitro*. Bacterial expressed GST-DUB3 WT and GST-DUB3 S41A fusion proteins were incubated with active CDK4 or CDK6 in the presence of [γ-$^{32}$P]ATP. Proteins were resolved by SDS–polyacrylamide gel electrophoresis; phosphorylated proteins were visualized with autoradiography. (**e**) FLAG-DUB3 WT or FLAG-DUB3 S41A was transfected in cells stably expressing DUB3 shRNA. Cell lysates were subjected to immunoprecipitation with anti-FLAG antibody and the phosphorylation of Ser41 in DUB3 was then examined. (**f**) Cells were transfected with indicated plasmids and were treated with vehicle, or CDK4/6 inhibitor (PD0332991). Cell lysates were subjected to immunoprecipitation with anti-FLAG antibody and the phosphorylation of Ser41 was examined. (**g**) Cells were transfected with indicated plasmids and cell lysates were subjected to immunoprecipitation with anti-FLAG antibody and western blot was performed. (**h**) MDA-MB-231 cells stably expressing indicated shRNAs were generated and western blot was performed with the indicated antibodies. (**i**) MDA-MB-231 cells stably expressing DUB3 shRNA were transfected with the WT, S41D or S41A mutant FLAG-DUB3 and treated with either vehicle or PD0332991. Western blot was performed with the indicated antibodies.

results establish that CDK4/6-mediated phosphorylation of DUB3 is important for DUB3 activity and SNAIL1 stability.

Overall, our study demonstrates that the CDK4/6–DUB3 axis functions as an important regulatory mechanism of breast cancer metastasis and provides a potential therapeutic approach in the management of breast cancer metastasis (Fig. 6h).

## Discussion

Despite advances in diagnosis and treatment, cancer metastasis is the most common cause of human cancer-related deaths and remains poorly understood. Metastasis occurs through a complex multistep process and requires the concerted action of many genes and signalling pathways[7,8]. In the past 20 years, accumulated experimental and clinical studies have demonstrated the crucial role of EMT in metastasis[8,9,17,18,45–49]. Thus, identification and characterization of the genes that regulate metastasis and the associated molecular mechanisms will lead to the development of new markers and potential therapeutic targets for cancer metastasis.

The DUB3 has been reported to regulate DNA damage response by controlling H2AX ubiquitination[50]. Moreover, overexpression

DUB3 has oncogenic potential by stabilizing the Cdc25A protein phosphatase in a subset of human breast cancers and couples G1/S checkpoint to pluripotency in mouse embryonic stem cells[51,52]. Here our studies indicate that DUB3 is crucial to induce EMT through the stabilization of SNAIL1 protein in breast cancer. In a subset of human breast cancer cell lines and patient tumour samples, the status of DUB3 is correlated with SNAIL1 expression (Fig. 3; Supplementary Fig. 5). Interestingly, elevated DUB3 and SNAIL1 expression are only detected in basal-like type but not luminal-type breast cancer cell lines. Furthermore, the correlation between DUB3 and SNAIL1 only exists in metastatic breast carcinoma but not in non-metastatic breast carcinoma (Supplementary Fig. 5). Thus, it will be important to elucidate how DUB3 expression is under regulatory control in breast cancer metastasis. Curiously, the DUB3 gene, a part of the highly polymorphic RS447 megasatellite sequence, is induced by cytokines interleukin (IL-)4 and IL-6 in certain mammalian cells[53,54]. IL-6 has been demonstrated to repress E-cadherin expression, promote EMT, invasion and metastasis in several different cancer types including breast cancer and colorectal cancer[55,56]. Given the link between cytokines and cancer metastasis, it will be intriguing to determine the regulatory

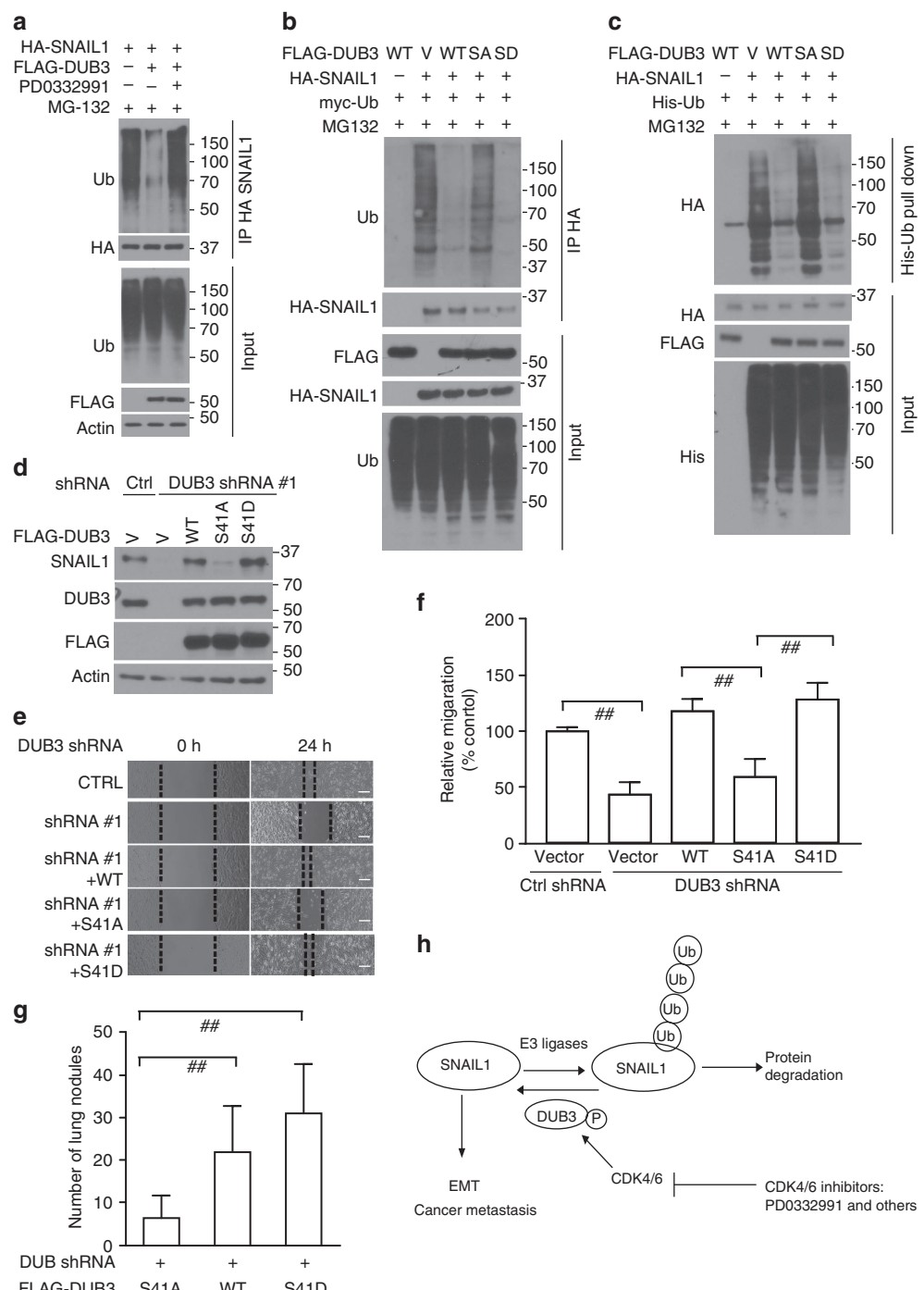

**Figure 6 | Phosphorylation of Ser41 regulates DUB3 activity.** (**a**) MDA-MB-231 cells were cotransfected with indicated plasmids and treated with vehicle or PD0332991. Cell lysates were boiled and immunoprecipitated with HA beads and immunoblotted as indicated. (**b**) Cells were transfected with indicated plasmids and the polyubiquitylated SNAIL1 protein was examined. (**c**) Cells were cotransfected with indicated plasmids and Ni-NTA beads were used to pull down His-tagged ubiquitin, and the polyubiquitylated SNAIL1 protein was examined. (**d**) Cells were transfected with indicated plasmids and western blot was performed as indicated. (**e,f**) The migratory ability of cells as in **d** was analysed by wound-healing assay. Scale bars, 100 μm. Results are quantified in **f**. The results represent the means ± s.d. of three independent experiments. $^{\#\#}P < 0.01$. (**g**) Cells stably expressing DUB3 shRNA were transfected with the WT, S41D or S41A mutant FLAG-DUB3 and one million cells from **c** were injected into the lateral tail vein of immunodeficient mice ($n = 8$). After 12 weeks, mice were killed and the development of lung metastases was determined by counting the number of metastatic lung nodules. Data are represented as mean ± s.d. Statistical analyses were performed with the Student's $t$-test. $^{\#\#}P < 0.01$. (**h**) The working model to illustrate that CDK4/6 phosphorylation-dependent activation of DUB3 regulates epithelial–mesenchymal transition and metastasis through SNAIL1.

mechanisms of DUB3 activation or expression in the cytokine context. SNAIL1 is also stabilized by inflammatory cytokines including IL-6 and tumour necrosis factor-α, in which activation of the nuclear factor-κB pathway is required to block the ubiquitination and degradation of SNAIL1 (ref. 43). Thus, the potential involvement of DUB3-mediated stabilization of SNAIL1

in inflammation-induced breast cancer metastasis is an interesting avenue that warrants further investigation.

Interestingly, we found that DUB3 is under regulatory control of CDK4/6 at the post-translational level. We demonstrate that phosphorylation of DUB3 on Ser41 is critical for activation of the enzymatic function. S41A mutant decreases DUB3 catalytic activity towards the fluorogenic substrate, Ub-AMC. Ser41 is located in an unstructured region of the protein, which is outside of the ubiquitin hydrolase domain. There are several reports implicating the regulatory phosphorylation of DUBs[57,58]. These post-translational modifications may activate or modulate regulatory subunits, which can control DUB activity and possibly substrate specificity. In addition, the phosphorylation modification of DUBs may trigger structural changes and affect ubiquitin recognition. Further detailed structural studies on DUB3 should be carried out to better understand the role of phosphorylation on its catalytic activity. Although we show that CDK4/6-mediated Ser41 phosphorylation is essential for DUB3 activation, phosphorylation of the other potential sites should be further investigated.

The mechanisms controlling metastasis can be regulated independently from primary tumour development. Recent studies using CDK4/6 inhibitors establish these cell cycle kinases as anti-cancer drug targets[59,60]. More recently, the Food and Drug Administration granted accelerated approval to the specific inhibitor of CDK 4/6, palbociclib (PD0332991), for use in combination with letrozole for the treatment of postmenopausal women with ER-positive, HER2-negative advanced breast cancer[39,40]. TNBCs usually grow the fastest and are more metastatic[36]. Our study reveals a potential use of CDK4/6 inhibitor in the treatment of TNBC metastasis. Consistent with previous studies, PD0332991 does not affect the growth of the primary tumour derived from human TNBC samples. Instead, PD0332991 induces the inactivation of DUB3, destablization of SNAIL1 protein and decrease in cell migration, thereby reducing metastasis in xenograft models, both from a breast cancer patient and a TNBC cell line. Given the facts that TNBC is particularly aggressive and more likely to metastasize, our study might introduce a new paradigm in the treatment of TNBC metastasis using CDK4/6 inhibitor. However, we only tested two TNBC breast cancer models here. The effect of CDK4/6 inhibitor in more TNBC models and other cancers needs to be further tested.

## Methods

**Cell culture, plasmids and antibodies.** 293T, MDA-MB-231, BT-549, MCF-7, T47D cells and other lines were purchased from American Type Culture Collection. Identities of all cell lines were confirmed by the medical genome facility at Mayo Clinic in Rochester, Minnesota, using short tandem repeat profiling upon receipt. Periodic Hoechst 33258 staining assays in these cells were performed to detect mycoplasma contamination.

DUB3 and SNAIL1 were cloned into pIRES-EGFP, pCMV-HA, pLV.3-FLAG, pGEX4T-1 and PET28A vectors. All site mutants were generated by site-directed mutagenesis (Stratagene) and verified by sequencing. DUB3, SNAIL1, CDK4 and CDK6 shRNAs were purchased from Sigma-Aldrich Co. The shRNA targeting sequences for CDK4 and CDK6 shRNAs are 5′-GAGATTACTTTGCTGCCTTA A-3′ and 5′-CAGATGTTGATCAACTAGGAA-3′, respectively. The sequences for SNAIL1 shRNA is 5′-CCAGGCTCGAAAGGCCTTCAA-3′. The sequences for DUB3 shRNAs are 5′-CACAAGCAGGtAGATCATCAC-3′ and 5′-GCAGGAA GATGCCCATGAATT-3′.

Antibodies against SNAIL1 (3895, dilution: 1:500), E-cadherin (14472, dilution: 1:1,000), N-cadherin (14215, dilution: 1:1,000), Vimentin (5741, dilution: 1:1,000), CDK substrate antibody (9477, dilution: 1:500) and Rb (9309, dilution: 1:1,000) were purchased from Cell Signaling Technology, Inc. Anti-FLAG (m2, dilution: 1:1,000), anti-HA (H3663, dilution: 1:1,000), anti-Myc (SAB4700447, dilution: 1:1,000) and anti-β-actin (A1978, dilution: 1:5,000) antibodies were purchased from Sigma-Aldrich Co. Ubiquitin, CDK4 (sc-23896, dilution: 1:500) and CDK6 (sc-7961, dilution: 1:500) antibodies were purchased from Santa Cruz Biotechnology, Inc. Antibodies used for immunohistochemistry against SNAIL1 (ab135708, dilution: 1:50) and DUB3 (ab129931, dilution: 1:50) were purchased

from Abcam. Rabbit anti-pSer41 (dilution: 1:100) was generated by immunizing rabbits with phospho-peptide, and then affinity-purified. Western blot was performed by using antibodies listed above. Uncropped scans of western blots were presented in Supplementary Figs 9–11.

**Patient-derived xenograft model.** A patient-derived xenograft model (HCI001) was generated from an oestrogen receptor-negative, progesterone receptor-negative, human epidermal growth factor receptor 2-negative $(ER^-/PR^-/HER2^-)$ high-grade invasive ductal carcinoma by the Breast Cancer Genome-Guided Therapy study (BEAUTY) in Mayo Clinic (A17713). Metastasis in immunodeficient mice implanted with human breast tumour biopsy sample HCI001 was found in the liver, lung as well as ovary reflecting the metastatic pattern in the donor patient. In brief, freshly operated human breast tumour biopsy samples were implanted in the flank of female immunodeficient NSG (NOD.Cg-Prkdcscid Il2rgtm1Wjl/SzJ) mice. When tumour size reached 1,000 mm³, mice were killed and tumour fragments (3 mm³) were re-transplanted into mammary fat pads of additional mice. Passage 3 tumours were used to test the effect of PD0332991 on metastasis. When primary tumours reached 100–150 mm³, mice were divided into two groups by stratified randomization and treated either with saline ($n=8$) or PD0332991 (150 mg kg⁻¹ daily, PO (oral administration), $n=8$) for an additional 5 weeks. Tumour volumes were measured once per week. After mice were killed, and lung and liver metastatic nodules were examined macroscopically or detected in paraffin-embedded sections stained with haematoxylin–eosin. Data were analysed using Student's $t$-test. A $P$ value $< 0.05$ was considered significant. Mice were subjected to euthanasia if they displayed pain or distress, such as lethargy, lying down, not eating or drinking, weight loss $> 10\%$ body weight or difficulty breathing. According to the blinding procedures, two people as a group performed all the mice experiments. One person performed the experiments and another one totally blinded to the experiment group measured the tumour volume and weight, number of lung and liver nodules, and analysed the data.

**Denaturing Ni-NTA pulldown.** Transiently transfected or virus infected cells were collected and pellets were washed once in PBS. Cells were lysed in 8 M urea, 0.1 M NaH₂PO₄, 300 mM NaCl and 0.01 M Tris (pH 8.0). Lysates were briefly sonicated to shear DNA and incubated with Ni-NTA agarose beads (Qiagen) for 1–2 h at room temperature. Beads were washed five times with 8 M urea, 0.1 M NaH₂PO₄, 300 mM NaCl and 0.01 M Tris (pH 8.0). Input and beads were boiled in loading buffer and subjected to SDS–polyacrylamide gel electrophoresis and immunoblotting.

**Denaturing immunoprecipitation for ubiquitination.** The cells were lysed in 100 μl 62.5 mM Tris-HCl (PH 6.8), 2% SDS, 10% glycerol, 20 mM NEM and 1 mM iodoacetamide, boiled for 15 min, diluted 10 times with NETN buffer containing protease inhibitors, 20 mM NEM and 1 mM iodoacetamide and centrifuged to remove cell debris. The cell extracts were subjected to immunoprecipitation with the indicated antibodies, and blotted as indicated.

**Immunofluorescence staining.** Cells were seeded onto glass coverslips for the experiment. Cells were washed with PBS, fixed with 4% formaldehyde for 10 min, permeabilized with 0.1% Triton X-100 for 5 min, blocked with 5% goat serum for 1 h, incubated with primary antibodies for 1 h and then with secondary antibodies. The antibodies used in the immunofluorescence staining were E-cadherin (1:200; Cell Signaling) and vimentin (1:500; Cell Signaling). Localization of E-cadherin and vimentin were visualized by confocal microscopy.

**Statistics.** For cell migration, invasion and proliferation experiments, data are represented as the mean ± s.d. of three independent experiments. In the animal study, data are represented as the mean ± s.d. of eight mice. Statistical analyses were performed with the Student's $t$-test, Fisher exact test, analsysi of variance or $\chi^2$-test. Statistical significance is represented in figures by: #$P < 0.05$; ##$P < 0.01$.

**Data availability.** All data generated or analysed during this study are available within the article and Supplementary Files, or available from the authors upon request.

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

## Acknowledgements

This work was supported by the National Basic Research Program of China (973 Program, grant no. 2013CB530700), National Natural Science Foundation of China (31270806, 81322031, 81572770 and 31371367), Mayo Clinic Breast Cancer SPORE (P50CA116201) and National Institutes of Health grants (CA203971, CA130996, CA189666 and CA203561). Breast Cancer Genome-Guided Therapy study (BEAUTY) was supported by Center for Individualized Medicine of Mayo Clinic.

## Author contributions

T.L. and Z.L. designed the experiments and analysed results. T.L. carried out the experiments and wrote the manuscript. J.Y., L.Z, J.Z. assisted with the analysis of IHC data. J.Y., B.Q. and P.Y. assisted in the animal experiment. M.D. and H.Z. assisted in *in vitro* kinase assay and *in vitro* deubiquitinase enzymatic assay. M.D., S.B.L., J.J.K., and J.L. assisted with the analysis of data; J.Y., L.W, J.B, M.G and Z.L. supervised the research. All authors discussed the results and commented on the manuscript.

**Additional information**

**Competing financial interests:** The authors declare no competing financial interests.

