## [Peer Review File · Nature Communications]

Reviewer #1 (Remarks to the Author)

In this study, the author demonstrated that inhibition of CDK4/6 blocked metastasis of TNBC, but not tumor growth, though inhibiting CDK4/6 mediated activation of DUB3, which is important for the stability of EMT factor SNAIL1. This finding revealed an important deubiquitination event for SNAIL and may have a potential therapeutic implication for TNBC metastasis. Overall, the study is very novel and solid. However, there are some technical concerns need to be addressed.

1. It is crucial to use CDK4/6 knockdown to confirm the effect of PD0332991 on SNAIL stability.
 2. Fig. 1d and 1g. Error bar and p value are absent in these figures.
 - Fig. 1k. A line graph with at least 3 independent experiments is required for the half life study.
 3. Fig. 1j. The author should quantify the results.
 4. Fig. 1l. It would be better to perform an endogenous ubiquitination assay to demonstrate that drug treatment induced SNAIL ubiquitination. K48R/K63R mutant ubiquitin construct and/or antibody specifically recognizing K48 or K63-linked poly-ubiquitination chain should be used to reveal which polyubiquitin linkage for SNAIL.
 5. The author should show the effect of DUB3 knockdown on the endogenous ubiquitination of SNAIL.
 6. Fig. 7a. A control with HA-SNAIL, but without FLAG-DUB3 should be included.
 7. Fig. 7c. Flag blot is missing.
 8. It would be critical to determine whether Dub3 regulates cancer metastasis acting through enhancing Snail stability".
- The authors should conduct the experiments to determine whether Snail knockdown reduces Dub3-mediated cancer metastasis and/or Snail restoration rescues cancer metastasis phenotypes upon Dub3 silencing.

Reviewer #2 (Remarks to the Author)

In this manuscript, Liu and coauthors report the novel finding of DUB3 as a phosphorylation target of CDK4/CDK6 that results in SNAIL1 deubiquitination and stabilization. They show that specific CDK4/6 inhibition by palbociclib (PD0332991) reduces the tumorigenicity ability of basal breast cancer lines in vivo, claiming that this is due to the reversion of the EMT phenotype in these cell lines. They associate this effect of PD0332991 to the decreased in SNAIL1 stability due to increased ubiquitination and degradation. They further uncover and probe that DUB3 is the deubiquitinase responsible for stabilizing SNAIL1 in basal like breast cancer cell lines. Liu et al show that silencing DUB3 impairs cell migration, invasion and lung colonization in experimental metastasis assays in vivo. They claim that DUB3 and SNAIL1 expression is significantly correlated in metastatic melanoma. The authors map Ser41 as the DUB3 residue specifically phosphorylated by CDK4/6 and responsible for DUB3 activity towards SNAIL1. Moreover, they are able to rescue the ability of basal like breast cells to colonize lungs when depleted for DUB3 and ectopically expressing a shRNA resistant DUB3 phosphomimetic mutant.

MAJOR POINTS

The manuscript presented herein entails novel and exciting results which are supported by wide-ranging in vitro and in vivo experiments performed by Liu and coauthors. The data enclosed in this manuscript reflects a remarkable effort and all together I'd suggest publishing this work upon reviewing several important issues.

- 1) The sentence in lane 47 should be rephrased since even when the loss of E-cadherin is hallmark of EMT, several events should take place to acquire an invasive phenotype.
- 2) Data regarding 1 PDX does not provide strong evidence, it'd be nice to show results from at least 2 PDX or consider mentioning them in supplementary.
- 3) As a general comment in the migration experiments performed it is necessary to quantify the relative migration in different experiments (n=3 at least) and show the results in a graph with

statistical analysis (Fig. 1a; Fig. 4d; Fig. 7e).

4) It is also intriguing the different SNAIL1 protein levels shown in different blots for the same cells as seen in Fig. 1i vs Fig. 1j (lanes corresponding to not treated cells).

5) Regarding the results shown in Fig. 2 it might be worth demonstrating DUB3 and SNAIL1 interaction in BT549 cells since the authors claim this binding is important in basal like cancer cells. CBS is not addressed in the main text and it is depicted in different blots.

6) The authors should show SNAIL1 mRNA measured by qPCR instead of RT-PCR since Fig. 3c is not clear. The CHX experiment in Fig. 3f should show SNAIL1 half-life since in the Ctrl cells SNAIL1 protein levels do not decrease after 1.5h.

7) DUB3 regulation of EMT is not properly addressed. The authors claim that DUB3 expression in luminal breast cancer cells promotes an EMT "converting luminal cells into basal-like phenotypes". To properly demonstrate this, the authors should check the status of other basal markers such as ER, EGFR and CK5. Moreover, the IF images depicted in Fig. 4b are not informative, E-cadherin is misslocalized. The experiment shown in Fig. 4f is not properly explained within the text, and only in methods. It should be stated herein that the cells are tail-vein injected.

Would the authors expect a difference in primary tumor growth? Discuss please.

8) The results presented in lines 182-186 are not properly introduced. The authors perform experiments in mouse melanoma cells. Do they claim then that DUB3 is also important in other cancer settings besides breast? Is DUB3 action in melanoma mediated by SNAIL1 stabilization which is not expressed in these cells? Please comment these results and discuss their relationship to the main message presented in the manuscript.

9) I have serious concerns regarding Fig. 5. First of all, the tables shown are not clear and the authors should make an effort to concisely present this information stressing the important findings and the authors' conclusion which is not clear to me from the data presented herein. Regarding the IHC, the images shown are quite unusual and I'd recommend displaying more images from both DUB3 and SNAIL1 IHC results. Moreover, the data regarding the type of carcinomas are vague. DUB3 and SNAIL1 expression should probably associate differently to subsets of breast carcinomas, luminal, basal.

10) In Fig. 7e it is written vehicle and PD0332991 below the panels, which is not mentioned in the text. Indeed the results presented would not fit easily in the presence of CDK4/6 inhibitor.

An experiment recommended to undoubtedly conclude that PD0332991 actions are mediated through CDK4/6-DUB3-SNAIL1 axis is to confirm in shDUB3 silenced cells incubated with PD0332991 that DUB3 SD ectopic expression rescues SNAIL1 stabilization whereas DUB3 SA not. Finally I'd suggest to rephrase in Discussion "our studies indicate that DUB3 is crucial to induce EMT through the stabilization of SNAIL1 protein" since the authors can conclude from their results that DUB3 is clearly involved in SNAIL1 stabilization which affects E-cadherin expression in the breast cancer cells lines they tested. But they would need to extend their results to other cellular settings to conclude this is a general DUB3 action.

MINOR POINTS

I recommend general reviewing of English writing since some sentences are not properly written.

Line 81: references 37 and 38 are not appropriate herein.

Line 100: the dramatically decrease in Vimentin is not seen in the figure 1i.

Line 239: explain succinctly what Ub-AMC.

Reviewer #3 (Remarks to the Author)

In this manuscript, the authors investigate the effect of the CDK4/6-inhibitor on formation of metastasis based on the triple-negative breast cancer cell line model. They identify DUB3 as a new substrate of CDK4/6. Phosphorylation of DUB3 at serine 41 leads to activation and as a result SNAIL1 is stabilized, which affects the epithelial-mesenchymal transition (EMT). Interestingly, the CDK4/6 inhibitor does not affect cell proliferation but rather cell migration, which reduces metastasis at least in an in vitro system.

Overall, this is an interesting story that is solely based on work in cell lines, which is its weakest

point. Nevertheless, the finding that DUB3 is a substrate of CDK4/6 and its consequences are of broad interest.

In addition, there are a number of shortcomings and issues that need to be addressed:

1. All the experiments presented are done in cell lines in vitro. Even the Xenograft and PDX models are not in vivo models of cancer. The "PDX" is actually not truly PDX since the breast tumor material was implanted in the flank and not in fat pad of the mouse breast. Therefore, one has to question the in vivo relevance of the presented findings. Even the staining in Fig.5 are not really convincing since the total DUB3 levels are not relevant - one would need to know if DUB3 is phosphorylated of S41 or not.
2. The description of the results in cell lines is fine but Fig.3c needs to be replaced by qPCR data.
3. In Fig.6f it is shown that Roscovitine is almost as effective as PD0322991. Since Roscovitine does not really inhibit CDK4/6, this result suggests that CDK2 and CDK1 are also involved. Therefore, the authors need to test if CDK2 and/or CDK1 can phosphorylate DUB3 in Fig.6d. In addition, for all reactions the authors need to include DUB3-S41A as a negative control.
4. Fig.7g the number of lung nodules are displayed for S41D and S41A. The authors also need to include a wild type DUB3 control (in the presence of DUB shRNA).
5. On line 74, PD0332991 is referred to as "highly specific". I am not sure that is really true with reports appearing that this inhibitor may affect proteins other than CDK4/6. I would suggest to phrase this more carefully.
6. This manuscript needs a lot of language editing (line 138/139, 173, 176, 201, 222, 228, 238, 239, 251, 269, and many, many more).
7. Fig.S1f: what is the purpose of showing a WB if there is no positive control (CDK substrate ab)?

Reviewers' comments:

Reviewer #1 (Remarks to the Author): Expert in ubiquitination and cancer

In this study, the author demonstrated that inhibition of CDK4/6 blocked metastasis of TNBC, but not tumor growth, though inhibiting CDK4/6 mediated activation of DUB3, which is important for the stability of EMT factor SNAIL1. This finding revealed an important deubiquitination event for SNAIL and may have a potential therapeutic implication for TNBC metastasis. Overall, the study is very novel and solid. However, there are some technical concerns need to be addressed.

Response: We thank the reviewer for the positive and constructive comments.

We performed experiments and revised the manuscript according to the reviewer's comments.

1. It is crucial to use CDK4/6 knockdown to confirm the effect of PD0332991 on SNAIL stability.

According to the reviewer's suggestion, we knockdown CDK4, CDK6 alone or together in MDA-MB-231 cells and SNAIL1 protein level was measured. As shown in Figure 6h, depletion of CDK4 or CDK6 alone in cells only partially decreased SNAIL1, while double knockdown of CDK4 and 6 decreased SNAIL1 dramatically.

2. Fig. 1d and 1g. Error bar and p value are absent in these figures.

Thanks for the reviewer's reminder. We added error bar and p value.

Fig. 1k. A line graph with at least 3 independent experiments is required for the half life study.

We performed 3 independent experiments and the half life of SNAIL1 in cells treated with Vehicle or PD0332991 was calculated (Figure 1l-m). We found that PD0332991 treatment dramatically decreased SNAIL1 stability.

3. Fig. 1j. The author should quantify the results.

According to reviewer's suggestion, we quantified results from 3 independent experiments in figure 1k. We found that PD0332991 treatment significantly decreased the SNAIL1 protein level in cells, and this decrease was rescued by MG132 treatment.

4. Fig. 1l. It would be better to perform an endogenous ubiquitination assay to demonstrate that drug treatment induced SNAIL ubiquitination. K48R/K63R mutant ubiquitin construct and/or antibody specifically recognizing K48 or K63-linked poly-ubiquitination chain should be used to reveal which polyubiquitin linkage for SNAIL.

We treated MDA-MB-231 cells with either vehicle or PD0332991. Endogenous SNAIL1 were immunoprecipitated and its ubiquitination level was detected (Figure 1o). In addition, Ubiquitin WT, K48 only, K63 only, K48R/K63R mutant was transfected in cells (Supplementary Fig.3b). We found that SNAIL1 was ubiquitinated through both K48 and K63-specific chains. However, DUB3 regulates only K48, but not K63 ubiquitin chains (Supplementary Fig.3c-d).

5. The author should show the effect of DUB3 knockdown on the endogenous ubiquitination of SNAIL.

We transfected control or DUB3 specific shRNAs in MDA-MB-231 cells and endogenous SNAIL1 were immunoprecipitated and its ubiquitination level was detected (Supplementary Fig.3a). We found that depletion of DUB3 significantly increased endogenous SNAIL1 ubiquitination.

6. Fig. 7a. A control with HA-SNAIL, but without FLAG-DUB3 should be included.

We performed this experiment and included the control as the reviewer suggested (Figure 7a).

7. Fig. 7c. Flag blot is missing.

We included Flag blot in Figure 7c as the reviewer suggested.

8. It would be critical to determine whether Dub3 regulates cancer metastasis acting through enhancing Snail stability". The authors should conduct the experiments to determine whether Snail knockdown reduces Dub3-mediated cancer metastasis and/or Snail restoration rescues cancer metastasis phenotypes upon Dub3 silencing.

We performed additional xenograft metastasis experiments and examined whether DUB3 regulated breast cancer metastasis through SNAIL1. We found that knockdown of DUB3 expression in MDA-MB-231 cells dramatically suppressed lung colonization, while restoration of SNAIL1 in these DUB3-depleted cells significantly increased lung colonization (Supplementary Fig. 4f-h). These results demonstrate that DUB3 is critical for TNBC metastasis through stabilizing SNAIL1.

Reviewer #2 (Remarks to the Author): Expert in breast cancer, EMT and metastasis

In this manuscript, Liu and coauthors report the novel finding of DUB3 as a phosphorylation target of CDK4/CDK6 that results in SNAIL1 deubiquitination and stabilization. They show that specific CDK4/6 inhibition by palbociclib (PD0332991) reduces the tumorigenicity ability of basal breast cancer lines in vivo, claiming that this is due to the reversion of the EMT phenotype in these cell lines. They associate this effect of PD0332991 to the decreased in SNAIL1 stability due to increased ubiquitination and degradation. They further uncover and probe that DUB3 is the deubiquitinase responsible for stabilizing SNAIL1 in basal like breast cancer cell lines. Liu et al show that silencing DUB3 impairs cell migration, invasion and lung colonization in experimental metastasis assays in vivo. They claim that DUB3 and SNAIL1 expression is significantly correlated in metastatic melanoma. The authors map Ser41 as the DUB3 residue specifically phosphorylated by CDK4/6 and responsible for DUB3 activity towards SNAIL1. Moreover, they are able to rescue the ability of basal like breast cells to colonize lungs when depleted for DUB3 and ectopically expressing a shRNA resistant DUB3 phosphomimetic mutant.

MAJOR POINTS

The manuscript presented herein entails novel and exciting results which are supported by wide-ranging *in vitro* and *in vivo* experiments performed by Liu and coauthors. The data enclosed in this manuscript reflects a remarkable effort and all together I'd suggest publishing this work upon reviewing several important issues.

Response: We thank the reviewer for the positive and constructive comments.

We performed experiments and revised the manuscript according to the reviewer's comments.

1) The sentence in lane 47 should be rephrased since even when the loss of E-cadherin is hallmark of EMT, several events should take place to acquire an invasive phenotype.

We rephrased the sentence according to the reviewer's comment. Please see Lane 47-50.

2) Data regarding 1 PDX does not provide strong evidence, it'd be nice to show results from at least 2 PDX or consider mentioning them in supplementary.

Thank you for the constructive comments.

So far, the Breast Cancer Genome-Guided Therapy study (BEAUTY) in Mayo Clinic generated only one metastatic PDX model, in which metastasis in immunodeficient mice implanted with human breast tumor biopsy sample reflects the metastatic pattern in the donor patient. We further investigated the effect of PD0332991 in a xenograft metastasis model using MDA-MB-231. MDA-MB-231 cells were injected into the mammary fat pad of immunodeficient mice. When tumors reached 400mm³ in size, we removed the primary tumors and treated these mice with either vehicle or PD0332991 for an additional 12 weeks. We found that the administration of PD0332991 dramatically suppressed lung colonization in these mice, as determined by metastatic lung nodule counting (Supplementary Fig. 1e-f). These results demonstrate that CDK4/6 inhibitor PD0332991 could inhibit MDA-MB-231 cell metastasis *in vivo*. Together with the PDX data, we conclude that CDK4/6 inhibitor PD0332991 inhibits breast cancer metastasis *in vivo*.

3) As a general comment in the migration experiments performed it is necessary to quantify the relative migration in different experiments (n=3 at least) and show the results in a graph with statistical analysis (Fig. 1a; Fig. 4d; Fig. 7e).

We quantified the relative migration activity and presented the graphic results in the related figures (Fig.1a ; Fig4d-Fig.4e; Fig. 7e-Fig.7f).

4) It is also intriguing the different SNAIL1 protein levels shown in different blots for the same cells as seen in Fig. 1i vs Fig. 1j (lanes corresponding to not treated cells).

This is probably due to different exposure. We repeated the experiment in Fig. 1j to get better blots

5) Regarding the results shown in Fig. 2 it might be worth demonstrating DUB3 and SNAIL1 interaction in BT549 cells since the authors claim this binding is important in basal like cancer cells. CBS is not addressed in the main text and it is depicted in different blots.

According to reviewer's comments, we performed the experiment and the interaction between DUB3 and SNAIL1 was detected by Co-immunoprecipitation in BT549 cells (Supplementary Fig 2e-f). CBS represents Coomassie Blue Staining and we now described it in the text.

6) The authors should show SNAIL1 mRNA measured by qPCR instead of RT-PCR since Fig. 3c is not clear. The CHX experiment in Fig. 3f should show SNAIL1 half-life since in the Ctrl cells SNAIL1 protein levels do not decrease after 1.5h.

According to the reviewer's suggestion, SNAIL1 mRNA levels in cell transfected with control and DUB3 shRNAs were measured by q-PCR (Figure 3c). For CHX assay, we performed 3 independent experiments and the half life of SNAIL1 in cells was quantified (Figure 3f-g). The half life of SNAIL1 was dropped from 1.5 h to 0.25 h after we depleted DUB3 in cells.

7) DUB3 regulation of EMT is not properly addressed. The authors claim that DUB3 expression in luminal breast cancer cells promotes an EMT "converting luminal cells into basal-like phenotypes". To properly demonstrate this, the authors should check the status of other basal markers such as ER, EGFR and CK5.

Thanks for the constructive comments. We checked the status of luminal marker (ER, FoxA1) and Basal marker (EGFR and CaV1) (Figure 4a). We found that expression of WT DUB3 resulted in decreased expression of luminal markers and increased basal markers, which is consistent with the conversion of luminal into basal-like phenotypes (Fig. 4a). Furthermore, the effect of DUB3 overexpression on EMT was not observed in cells stably expressing SNAIL1 shRNA, suggesting that DUB3 functions in promoting basal-like phenotype conversion through SNAIL1.

Moreover, the IF images depicted in Fig. 4b are not informative, E-cadherin is misslocalized.

As shown in Fig.4b, The immunofluorescence experiment was repeated and localization of E-cadherin and Vimentin were visualized by confocal microscopy.

The experiment shown in Fig. 4f is not properly explained within the text, and only in methods. It should be stated herein that the cells are tail-vein injected.

We revised the Figure Legends (Now is Fig. 4g) accordingly.

Would the authors expect a difference in primary tumor growth? Discuss please.

The cell proliferation rate was measured in cell stably expressing control and DUB3 shRNAs. No difference of cell proliferation was observed (Supplementary Fig.4a).

Furthermore, we injected these cells into the mammary fat pad of female immunodeficient mice and the tumor volume was measured. No significant difference of tumor volume was

observed (Supplementary Fig. 4b). All these *in vitro* and *in vivo* results suggested that DUB3 does not regulate primary tumor growth.

8) The results presented in lines 182-186 are not properly introduced. The authors perform experiments in mouse melanoma cells. Do they claim then that DUB3 is also important in other cancer settings besides breast? Is DUB3 action in melanoma mediated by SNAIL1 stabilization which is not expressed in these cells? Please comment these results and discuss their relationship to the main message presented in the manuscript.

Our results suggest that DUB3 regulates SNAIL1 protein level in B16F10-SNAIL1⁺ melanoma cells. Furthermore, downregulation of DUB3 also decreased the cell migration in these cells, suggesting that DUB3 may regulate cell migration in melanoma cells. However, we don't have *in vivo* data yet. To be more focused on the function of CDK4/6-DUB3-SNAIL1 in TNBC metastasis, we deleted the mouse melanoma experiments from the text.

9) I have serious concerns regarding Fig. 5. First of all, the tables shown are not clear and the authors should make an effort to concisely present this information stressing the important findings and the authors' conclusion which is not clear to me from the data presented herein. Regarding the IHC, the images shown are quite unusual and I'd recommend displaying more images from both DUB3 and SNAIL1 IHC results. Moreover, the data regarding the type of carcinomas are vague. DUB3 and SNAIL1 expression should probably associate differently to subsets of breast carcinomas, luminal, basal.

Thanks for reviewer's suggestion. We re-stained the tissue microarray and re-made the tables to make it clearer. As shown in Figure 5b, high DUB3 protein expression positively correlated with metastatic carcinoma (Fig. 5a-b, $P < 0.0001$, $R = 0.347$). High SNAIL1 protein expression was also positively related with metastatic carcinoma (Fig. 5a&c, $P < 0.0001$, $R = 0.344$).). We also tested the possible correlation between DUB3 and SNAIL1 protein levels using the tissue microarray. Consistent with our hypothesis, high DUB3 levels positively correlated with high SNAIL1 in the metastatic carcinoma (Fig. 5d). The

above mentioned microarray did not have breast subtype information. To test the correlation with breast tissue subtypes, we used another breast tissue microarray and found both high DUB3 and high SNAIL1 protein expressions positively correlated with basal like breast cancer type (Supplementary Fig. 5 c-d). We also present more images from both DUB3 and SNAIL1 IHC results according to reviewer's suggestion (Fig. 5a, Supplementary Fig. 5 a-b).

10) In Fig. 7e it is written vehicle and PD0332991 below the panels, which is not mentioned in the text. Indeed the results presented would not fit easily in the presence of CDK4/6 inhibitor. An experiment recommended to undoubtedly conclude that PD0332991 actions are mediated through CDK4/6-DUB3-SNAIL1 axis is to confirm in shDUB3 silenced cells incubated with PD0332991 that DUB3 SD ectopic expression rescues SNAIL1 stabilization whereas DUB3 SA not.

The written "vehicle" and "PD0332991" was mistakenly attached during the organization of Figure 7. To assess whether PD0332991 actions are mediated through CDK4/6-DUB3-SNAIL1 axis, we expressed WT DUB3, S41A or S41D mutants in cells depleted of endogenous DUB3 and then treated these cells with either vehicle or PD0332991. SNAIL1 protein levels were measured. We found that the treatment of PD0332991 in cells reconstituted with WT DUB3 significantly decreased SNAIL1 protein level (Fig. 6i). SNAIL1 protein level was high in cells reconstituted with S41D and low in cells reconstituted with S41A. Importantly, PD0332991 treatment had no further effect on SNAIL1 expression in cells expressing S41A or S41D, suggesting that PD0332991 acts through the CDK4/6-DUB3-SNAIL axis.

Finally I'd suggest to rephrase in Discussion "our studies indicate that DUB3 is crucial to induce EMT through the stabilization of SNAIL1 protein" since the authors can conclude from their results that DUB3 is clearly involved in SNAIL1 stabilization which affects E-cadherin

expression in the breast cancer cells lines they tested. But they would need to extend their results to other cellular settings to conclude this is a general DUB3 action.

Thanks for reviewer's suggestion. We revised the discussion in the text. Please see lane 305.

MINOR POINTS

I recommend general reviewing of English writing since some sentences are not properly written.

We have proofread the manuscript carefully and corrected typos/grammar mistakes.

Line 81: references 37 and 38 are not appropriate herein.

We removed these two references from this sentence.

Line 100: the dramatically decrease in Vimentin is not seen in the figure 1i.

We rephrase the sentence in the text. Please see lane 105.

Line 239: explain succinctly what Ub-AMC.

We add some information of ub-AMC in the method. Please see lane 19-26 in supplementary information.

Reviewer #3 (Remarks to the Author): Expert in CDK and cancer

In this manuscript, the authors investigate the effect of the CDK4/6-inhibitor on formation of metastasis based on the triple-negative breast cancer cell line model. They identify DUB3 as a new substrate of CDK4/6. Phosphorylation of DYB3 at serine 41 leads to activation and as a result SNAIL1 is stabilized, which affects the epithelial-mesenchymal transition (EMT).

Interestingly, the CDK4/6 inhibitor does not affect cell proliferation but rather cell migration,

which reduces metastasis at least in an in vitro system.

Overall, this is an interesting story that is solely based on work in cell lines, which is its weakest point. Nevertheless, the finding that DUB3 is a substrate of CDK4/6 and its consequences are of broad interest.

Response: We thank the reviewer for the constructive comments. We performed experiments and revised the manuscript according to the reviewer's comments.

In addition, there are a number of shortcomings and issues that need to be addressed:

1. All the experiments presented are done in cell lines in vitro. Even the Xenograft and PDX models are not in vivo models of cancer. The "PDX" is actually not truly PDX since the breast tumor material was implanted in the flank and not in fat pad of the mouse breast. Therefore, one has to question the in vivo relevance of the presented findings. Even the staining in Fig.5 are not really convincing since the total DUB3 levels are not relevant - one would need to know if DUB3 is phosphorylated of S41 or not.

We understand the reviewer's concerns. According to reviewer's comment, we modified our xenograft metastasis model and further investigated PD0332991 function using this model. MDA-MB-231 cells were injected into the mammary fat pad of immunodeficient mice. When tumors reached 400mm³ in size, we removed the primary tumors and treated these mice with either vehicle or PD0332991 for additional 12 weeks. We found that the administration of PD0332991 dramatically suppressed lung colonization in these mice, as determined by metastatic lung nodule counting (Supplementary Fig. 1e-f). Although each model has its limitation, taking our data as a whole, using cell line or PDX through different approaches (tail vein injection, flank implantation, and mammary pad implantation), we conclude that the DUB3-SNAIL1 axis is critically involved in TNBC metastasis.

Although our phospho-S41 antibody is good for western blotting, it is not useful for IHC. We were not able to measure the potential correlation between phospho-S41 and SNAIL1 in the tissue microarray. We think that both DUB3 level and phosphorylation are important DUB activity in cells and for SNAIL stability, as simply overexpression or depletion of DUB3 affects SNAIL1 stability. Figure 5 indicate that DUB3 level positively correlates with SNAIL1 levels in the metastatic carcinoma (Fig. 5d), this is consistent with our model.

2. The description of the results in cell lines is fine but Fig.3c needs to be replaced by qPCR data.

SNAIL1 mRNA levels in cell transfected with control and DUB3 shRNAs were measured by q-PCR (Figure 3c).

3. In Fig.6f it is shown that Roscovitine is almost as effective as PD0322991. Since Roscovitine does not really inhibit CDK4/6, this result suggests that CDK2 and CDK1 are also involved. Therefore, the authors need to test if CDK2 and/or CDK1 can phosphorylate DUB3 in Fig.6d. In addition, for all reactions the authors need to include DUB3-S41A as a negative control.

Thanks for the reviewer's comments. We included GST-S41A as a negative control and tested whether CDK2 and CDK1 can phosphorylate DUB3. Indeed, CDK1, but not CDK2, could phosphorylate DUB3 *in vitro* although the intensity of p32 signal is weaker than CDK6 (Supplementary Fig. 7a). The potential function of CDK1 in regulating metastasis will be investigated in the future.

4. Fig.7g the number of lung nodules are displayed for S41D and S41A. The authors also need to include a wild type DUB3 control (in the presence of DUB shRNA).

We expressed WT DUB3, S41A or S41D mutants in MDA-MB-231 cells depleted of endogenous DUB3 and a xenograft metastasis experiment was performed. We found that cells expressing WT DUB3 and the S41D mutant exhibit stronger lung colonization than

those expressing the S41A mutant, as determined by lung nodules counting (Fig. 7g) and by intensity of bioluminescence (Supplementary Fig. 8a-b).

5. On line 74, PD0332991 is referred to as "highly specific". I am not sure that is really true with reports appearing that this inhibitor may affect proteins other than CDK4/6. I would suggest to phrase this more carefully.

We rephrased the sentence as suggested. Please see lane 73.

6. This manuscript needs a lot of language editing (line 138/139, 173, 176, 201, 222, 228, 238, 239, 251, 269, and many, many more).

We have proofread the manuscript carefully and corrected typos/grammar mistakes.

7. Fig.S1f: what is the purpose of showing a WB if there is no positive control (CDK substrate ab)?

We included Rb as positive control (Supplementary Fig 1h).

Reviewer #1 (Remarks to the Author)

The authors have adequately addressed all of my concerns. The study reveals the novel role of DUB3 in cancer metastasis through promoting Snail stability. The study is novel and well supported by experimental data. It is now suitable for publications at Nature Communications.

Reviewer #2 (Remarks to the Author)

CDK4/6 Dependent Activation of DUB3 Regulates Cancer Metastasis through SNAIL1

The authors have properly addressed the concerns raised initially by the reviewers. The manuscript has soundly improved upon revision, and the novel and interesting results are now supported by stronger evidence. The data enclosed reflect a remarkable effort and all together I'd recommend its publishing.

Nevertheless, I should mention that the authors overemphasize their conclusions regarding the molecular pathways linking CDK4/6 and DUB3 to TNBC since their models are limited to two basal-like breast cancer cell lines and one PDX. I'd suggest this should be somehow made clearer throughout the text and especially in the discussion.

Minor points

- Abstract: mechanistically might fit better than mechanically (line30)
- Line 114-115: the interaction between CDK4/6 and the E3 ligases is not expected in any case, the authors can eliminate this sentence
- Results presented in Figure 1o are not properly addressed in the text
- Line 326: mutant of this phosphorylation site should be rephrased
- Model Fig. 8: should be improved for better appreciation, cancer has an erratum
- Suppl. Fig. 8 is firstly mentioned in the discussion which seems odd

Reviewer #3 (Remarks to the Author)

The authors have revised their manuscript and overall it is clearly improved. The connection between CDK4/6-DUB3-SNAIL is interesting but ultimately may not be as simple as the authors describe here.

Nevertheless, there are a few shortcomings that absolutely need to be fixed before this manuscript can be published:

1. In several places, the authors mention over and over again "in vivo". This point was already mentioned in the first round of revisions. The authors have not done a single experiment "in vivo" but they are trying to sell their experiments as "in vivo" which is not acceptable. They have to remove all mentioning of "in vivo" from their manuscript.
2. Figure 5 is still a problem as was also pointed out by another reviewer. This figure is not important enough to be shown in the main part of the manuscript. Either it should be moved to Supplemental Materials or removed.
3. The language in this manuscript should be further improved.

Reviewer #1 (Remarks to the Author):

The authors have adequately addressed all of my concerns. The study reveals the novel role of DUB3 in cancer metastasis through promoting Snail stability. The study is novel and well supported by experimental data. It is now suitable for publications at Nature Communications.

Thanks for the reviewer's positive comments.

Reviewer #2 (Remarks to the Author):

CDK4/6 Dependent Activation of DUB3 Regulates Cancer Metastasis through SNAIL1

The authors have properly addressed the concerns raised initially by the reviewers. The manuscript has soundly improved upon revision, and the novel and interesting results are now supported by stronger evidence. The data enclosed reflect a remarkable effort and all together I'd recommend its publishing. Nevertheless, I should mention that the authors overemphasize their conclusions regarding the molecular pathways linking CDK4/6 and DUB3 to TNBC since their models are limited to two basal-like breast cancer cell lines and one PDX. I'd suggest this should be somehow made clearer throughout the text and especially in the discussion.

Thanks for the reviewer's positive comments. We made changes in the main text according to reviewer's comments.

Minor points

- Abstract: mechanistically might fit better than mechanically (line30)

The word was changed accordingly.

- Line 114-115: the interaction between CDK4/6 and the E3 ligases is not expected in any case, the authors can eliminate this sentence

This sentence was deleted from the text.

- Results presented in Figure 1o are not properly addressed in the text

We described the figure 1o in the main text in line 121: PD0332991 treatment increased the ubiquitination of SNAIL1 (Fig. 1n-o)

- Line 326: mutant of this phosphorylation site should be rephrased

We change mutant of this phosphorylation site to S41A mutant.

- Model Fig. 8: should be improved for better appreciation, cancer has an erratum

We corrected the mistake.

- Suppl. Fig. 8 is firstly mentioned in the discussion which seems odd

We mentioned this Supplementary Figure in the main text.

Reviewer #3 (Remarks to the Author):

The authors have revised their manuscript and overall it is clearly improved. The connection between CDK4/6-DUB3-SNAIL is interesting but ultimately may not be as simple as the authors describe here. Nevertheless, there are a few shortcomings that absolutely need to be fixed before this manuscript can be published:

Thanks for the reviewer's positive comments. We made changes in the main text according to reviewer's comments.

1. In several places, the authors mention over and over again "in vivo". This point was already mentioned in the first round of revisions. The authors have not done a single experiment "in vivo" but they are trying to sell their experiments as "in vivo" which is not acceptable. They have to remove all mentioning of "in vivo" from their manuscript.

We deleted all ""in vivo" in the main text according to the reviewer's suggestion.

2. Figure 5 is still a problem as was also pointed out by another reviewer. This figure is not important enough to be shown in the main part of the manuscript. Either it should be moved to Supplemental Materials or removed.

We move the Figure 5 to supplementary materials.

3. The language in this manuscript should be further improved.

We have manuscript proofread by native speakers as suggested.